# Multi-scale event synchronization analysis for unravelling climate processes: A wavelet-based approach

Ankit Agarwal[1, 2, 3,] Norbert Marwan[2], Maheswaran Rathinasamy[4], Bruno Merz[1, 3], Jürgen Kurths[1, 2]

[1]University of Potsdam, Institute of Earth and Environmental Science, Karl-Liebknecht-Strasse 24-25, 14476 Potsdam, Germany
[2]Potsdam Institute for Climate Impact Research, P.O. Box 60 12 03, 14412 Potsdam, Germany
[3]GFZ German Research Centre for Geosciences, Section 5.4: Hydrology, Telegrafenberg, Potsdam, Germany
[4]MVGR College of Engineering, Vizianagaram, India

*Correspondence to*: A. Agarwal (aagarwal@uni-potsdam.de)

**Abstract.** The temporal dynamics of climate processes are spread across different time scales and, as such, the study of these processes only at one selected time scale might not reveal the complete mechanisms and interactions within and between the (sub-) processes. For capturing the nonlinear interactions between climatic events, the method of event synchronization has found increasing attention recently. The main drawback with the present estimation of event synchronization is its restriction to analyse the time series at one reference time scale only. The study of event synchronization at multiple scales would be of great interest to comprehend the dynamics of the investigated climate processes. In this paper, wavelet based multi-scale event synchronization (MSES) method is proposed by combining the wavelet transform and event synchronization. Wavelets are used extensively to comprehend multi-scale processes and the dynamics of processes across various time scales. The proposed method allows the study of spatio-temporal patterns across different time scales. The method is tested on synthetic and real-world time series in order to check its replicability and applicability. The results indicate that MSES is able to capture relationships that exist between processes at different time scales.

**Keywords:** multi-scale, event synchronization, discrete wavelet transformation, significance test

## 1 Introduction

Synchronization is a wide-spread phenomenon that can be observed in numerous climate-related processes, such as synchronized climate changes of the north and south Polar Regions (Rial 2012), see-saw relationship between monsoon systems (Eroglu et al., 2016), or coherent fluctuations in flood activity across regions (Schmocker-Fackel and Naef 2010) and among El Niño and the Indian summer monsoon (Marun and Kurths 2005; Mokhov et al., 2011). Synchronous occurrences of climate-related events can be of great societal relevance. The occurrence of strong precipitation or extreme runoff, for instance, at many locations within a short time period may overtax the disaster management capabilities.

Various methods for studying synchronization are available, based on recurrences (Marwan et al., 2007; Donner et al., 2010; Arnhold et al., 1999; Le Van Quyen et al., 1999; Quiroga et al., 2000; Quiroga et al., 2002; Schiff et al., 1996), phase differences (Schiff et al., 1996, Rosenblum et al., 1997), or the quasi-simultaneous appearance of events [Tass et al., 1998; Stolbova et al., 2014; Malik et al., 2012; Rheinwalt et al., 2016). For the latter, the method of event synchronization (ES) has

received popularity owing to its simplicity, in particular within the fields of brain [Pfurtscheller and Silva 1999; Krause et al., 1996) and cardiovascular research (O'Connor et al., 2013), non-linear chaotic systems (Callahan et al., 1990), and climate sciences (Tass et al., 1998; Stolbova et al., 2014; Malik et al., 2012; Rheinwalt et al., 2016). ES has also been used to understand driver-response relationships, i.e. which process leads and possibly triggers another based on its asymmetric property. It has been shown that, for event-like data, ES delivers more robust results compared to classical measures such as

correlation or coherence functions which are limited by the assumption of linearity (Liang et al., 2016).

Particularly in climate sciences, ES has been successfully applied to capture driver-response relationships, time delays between spatially distributed processes, strength of synchronization, moisture source and rainfall propagation trajectories, and to determine typical spatio-temporal patterns in monsoon systems (Stolbova et al., 2014; Malik et al., 2012; Rheinwalt et al., 2016). Furthermore, extensions of the ES approach have been suggested to increase its robustness with respect to

boundary effects (Stolbova et al., 2014; Malik et al., 2012) and number of events (Rheinwalt et al., 2016).

Even though ES has been successfully used, it is yet limited by measuring the strength of the nonlinear relationship at only one given temporal scale, i.e. it does not consider relationships at and between different temporal scales. However, climate-related processes typically show variability at a range of scales. Synchronization and interaction can occur at different temporal scales, as localized features, and can even change with time (Rathinasamy et al., 2014; Herlau et al., 2012;

Steinhaeuser et al., 2012; Tsui 2015). Features at a certain time scale might be hidden while examining the process at a different scale. Also, some of the natural processes are complex due to the presence of scale emergent phenomena triggered by nonlinear dynamical generating processes, long-range spatial and long-memory temporal relationships (Barrat et al., 2008). In addition, single-scale measures, such as correlation and ES, are valid and meaningful only for stationary systems. For non-stationary systems, they may underestimate or overestimate the strength of the relationship (Rathinasamy et al.,

25 2014).

The wavelet transform can potentially convert a non-stationary time series into stationary components (Rathinasamy et al., 2014), and this can help in analysing non-stationary time series using the proposed method.

Therefore, the multi-scale analysis of climatic processes holds the promise to better understand the system dynamics that may be missed when analysing processes at one time scale only (Perra et al., 2012; Miritello et al., 2013). According to this

background, we propose a novel method, the Multi-Scale Event Synchronization (MSES) which integrates ES and wavelet approach in order to analyse synchronization between event time series at multiple temporal scales. To test the effectiveness of the proposed methodology, we apply it to several synthetic and real-world test cases.

The manuscript is organized as follows: Section 2 describes the proposed methodology and Section 3 introduces selected case studies. The results are discussed in Section 4. Conclusions are summarized in Section 5.

## 2 Methods

Here we describe the methodology for the proposed MSES approach. In this we combine two already well established approaches (DWT and ES) to analyse synchronization at multiple temporal scales. The following sub sections briefly introduce wavelets & ES and subsequently provide the mathematical framework for estimating MSES.

### 2.1 Discrete wavelet transform

Wavelet analysis has become an important method in spectral analysis due to its multi-resolution and localization capability both in time and frequency domain. A wavelet transform converts a function (or signal) into another form which makes certain features of the signal more amenable to study (Addison 2005). A wavelet $\psi(t)$ is a localized function which satisfies certain admissibility conditions. The wavelet transform $T_{a,b}(x)$ of a continuous function $x(t)$ can be defined as a simple convolution between $x(t)$ and dilated and translated versions of the mother wavelet $\psi(t)$:

$$T_{a,b}(x) = \int_{-\infty}^{\infty} x(t)\psi_{a,b}(t)dt, \tag{1}$$

where $a$ $and$ $b$ refer to the scale and location variables (real numbers) and $\psi_{a,b}$ is defined as

$$\psi_{a,b}(t) = \frac{1}{\sqrt{a}}\, \psi\left(\frac{t-b}{a}\right). \tag{2}$$

Depending on the way we sample the parameters $a$ $and$ $b$, we get either a continuous wavelet transform (CWT) or a discrete wavelet transform (DWT). A natural way to sample $a$ and $b$ is to use a logarithmic discretization of the scale and link this in turn to the size of steps taken between $b$ locations. This kind of discretization of the wavelet has the form

$$\psi_{\lambda,q}(t) = \frac{1}{\sqrt{a_o^\lambda}}\psi\left(\frac{t-qb_o a_o^\lambda}{a_o^\lambda}\right), \tag{3}$$

Where the integers $\lambda$ and q control the wavelet dilation and translation respectively; $a_o$ is a specified fixed dilation step parameter and $b_o > 0$ is the location parameter. The general choices of the discrete wavelet parameters $a_o$ and $b_o$ are 2 and 1, respectively. This is known as dyadic grid arrangement.

Using the dyadic grid wavelet, the DWT can be written as

$$T_{\lambda,q} = \int_{-\infty}^{\infty} x(t)\, \frac{1}{\sqrt{a_o^\lambda}}\, \psi(\frac{t-qb_o a_o^\lambda}{a_o^\lambda})\, dt \quad \text{Substituting } a_0 = 2 \; and \; b_o = 1; \text{ we get} \tag{4}$$

$$T_{\lambda,q} = \int_{-\infty}^{\infty} x(t).2^{-\lambda/2}\, \psi(2^{-\lambda}t - q)dt,$$

where $T_{\lambda,q}$ are the discrete wavelet transform values given on a scale-location grid index $\lambda$ and q. For the DWT, the values $T_{\lambda,q}$ are known as wavelet coefficients or *detail coefficients*.

The decomposition of the dyadic discrete wavelet is also associated with the scaling function $\phi_{\lambda,q}(t)$, (eq. 5) which represents the smoothing of the signal and has the same form as the wavelet, given by (Addison 2005)

$$\phi_{\lambda,q}(t) = 2^{-\frac{\lambda}{2}} \phi(2^{-\lambda}t - q)$$

(5)

The scaling function is orthonormal to the translation of itself, but not to the dilation of itself. $\phi_{\lambda,q}(t)$ can be convolved with the signal to produce *approximation coefficients* at a given scale as follows:

$$A_{\lambda,q} = \int_{-\infty}^{\infty} x(t)\phi_{\lambda,q}(t)dt$$

(6)

The approximation coefficients at a specific scale $\lambda$ are known as a discrete approximation of the signal at that scale. As proven in (Mallat 1989), the wavelet function and the scaling function form multi- resolution bases resulting in a pyramidal algorithm. The decomposition methodology is schematically shown in Figure 1.

In this study, to calculate the synchronization at multiple scales we only consider the approximation coefficients (not detail coefficients) at that particular scale because the aim is to separate the effects of time-localized features and high frequency

components from the signal.

For different $\lambda = 1,2,3$ ..., the approximation coefficients $A_\lambda$ correspond to the "coarse-grained" original signal after removal of the details at scales $\lambda$, $\lambda - 1, ... ,1$. In practical terms, considering a daily climatic time series at $\lambda = 0$, the time series represents the original observations. At $\lambda = 1, A_1$ represents the features beyond the 2-day scale (wavelet scale) which is obtained by extracting $T_1$ (2-day features) from the original time series. Similarly, at $\lambda = 3$, $A_3$ represents the climatic

variable beyond the 8-day scale and is obtained after removing $T_1$, $T_2$, $T_3$ (2, 4, 8 day features) from the original signal. In essence, $A_1$, $A_2$, $A_3,...$ represent the original signal at different time scales. The schematic plot explaining the procedure and relationship between signal, approximate component and detailed component has been shown in Fig.2.

For simplicity we denote the approximation coefficient $A_{\lambda,q}$ of the signal $x(t)$ at scale $\lambda$ as $x_\lambda$.

**2.2 Event synchronization**

To quantify the synchronous occurrence of events in different time series, we use the Event Synchronization (ES) method proposed by (Quiroga et al., 2002). ES can be used for any time series in which we can define events, such as single-neuron recordings, eptiform spikes in EEGs, heart beats, stock market crashes, or abrupt weather events, such as heavy rainfall events. However, ES is not limited to this definition of events. It could also be applied to time series which are pure event

time series (e.g. heart beats). In principle, when dealing with signals of different character, the events could be defined differently in each time series, since their common cause might manifest itself differently in each (Quiroga et al., 2002). ES has advantages over other time-delayed correlation techniques (e.g., Pearson lag correlation), as it allows us to study interrelations between series of non-Gaussian data, data with heavy tails, or using a dynamical (non-constant) time delay

(Tass et al., 1998; Stolbova et al., 2014). The latter refers to a time delay that is dynamically adjusted according to the two time series being compared, which allows for better adaptation to the region of interest. Furthermore, ES has been specifically designed to calculate nonlinear linkages between time series. Various modifications of ES have been proposed, such as solving the problems of boundary effects and bias due to an infinite number of events (Stolbova et al., 2014; Malik et al., 2012; Rheinwalt et al., 2016).

The modified algorithm proposed by (Stolbova et al., 2014; Malik et al., 2012; Rheinwalt et al., 2016) works as follows: An event occurs in the signals $x(t)$ and $y(t)$ at time $t_l^x$ and $t_m^y$, where $l = 1,2,3,4 \dots S_x$, $m = 1,2,3,4 \dots \dots S_y$, and $S_x$ & $S_y$ are the total number of events, respectively. In our study, we derive events from a more-or-less continuous time series by selecting all time steps with values above a threshold ($\alpha = 95^{th}$ percentile). These events in $x(t)$ and $y(t)$ are considered as synchronized when they occur within a time lag $\pm\tau_{lm}^{xy}$ which is defined as follows:

$$\tau_{lm}^{xy} = min\left\{t_{l+1}^x - t_l^x, t_l^x - t_{l-1}^x, t_{m+1}^y - t_m^y, t_m^y - t_{m-1}^y\right\}/2 \tag{7}$$

This definition of the time lag helps to separate independent events, as it is the minimum time between two succeeding events. Then we count the number of times C(x|y) an event occurs in $x(t)$ after it appears in $y(t)$ and vice versa (C(y|x)):

$$C(x|y) = \sum_{l=1}^{S_x} \sum_{m=1}^{S_y} J_{xy} \tag{8}$$

And

$$J_{xy} = \begin{cases} 1 \; if \;\; 0 < t_l^x - t_m^y < \tau_{lm}^{xy} \\ \dfrac{1}{2} if \; t_l^x = t_m^y \\ \quad 0 \, else, \end{cases} \tag{9}$$

$C(y|x)$ is calculated analogously but with exchanged $x$ and $y$. From these quantities we obtain the symmetric measure:

$$Q_{xy} = \frac{C(x|y) + C(y|x)}{\sqrt{(S_x - 2)(S_y - 2)}} \tag{10}$$

$Q_{xy}$ is a measure of strength of event synchronization between signal $x(t)$ and $y(t)$. It is normalized to $0 \leq Q_{xy} \leq 1$, with $Q_{xy} = 1$ for perfect synchronization (coincidence of extreme events) between signals $x(t)$ and $y(t)$.

Recalling Eq. 6, the scale-wise approximation at different scales $0, 1, 2, \dots, \lambda$ for any given time series x(t) is given by $x_\lambda = A_{\lambda,q}$. Where $x_\lambda$ represents the approximation coefficients of signal x(t) at scale $\lambda$. Now, for determining the synchronization between any two time series x(t) and y(t) at multiple scales, the event synchronization is estimated between the scaled versions of x(t) and y(t) for different $\lambda$ resulting in the multi-scale event synchronization (MSES). The normalized strength of MSES between the signals x(t) and y(t) at scale $\lambda$ is then defined as:

$$Q^{x_\lambda, y_\lambda} = \frac{C(x_\lambda | y_\lambda) + C(y_\lambda | x_\lambda)}{\sqrt{(S_{x_\lambda} - 2)(S_{y_\lambda} - 2)}} \tag{11}$$

$Q^{x_\lambda, y_\lambda} = 1$ for perfect synchronization, and $Q^{x_\lambda, y_\lambda} = 0$ suggests the absence of any synchronization at scale $\lambda$ between $x(t)$ and $y(t)$.

Figure 3 shows the stepwise methodology of Multiscale event synchronization.

## 2.3 Significance test for MSES

To evaluate the statistical significance of ES values, a surrogate test will be used (Rheinwalt et al., 2016). We randomly reshuffle each time series 100 times (an arbitrary number). Reshuffling is done without replacement because estimating the expected number of simultaneous events in independent time series is equivalent to the combinatorial problem of sampling without replacement (Rheinwalt et al., 2016). Then, for each pair of time series, we calculate the MSES values for the different scales. At each scale, the empirical test distribution of the 100 MSES values for the reshuffled time series is

compared to the MSES values of the original time series. Using a 1% significance level, we assume that synchronization cannot be explained by chance, if the MSES value at a certain scale of the original time series is larger than the 99[th] percentile of the test distribution.

## 3 Data and study design to test MSES

The proposed method is tested using synthetic and real-world data. The aim of these tests is to understand whether MSES is

advantageous, compared to ES, in understanding the system interaction and the scale-emerging natural processes.

## 3.1 Testing MSES with synthetic data

Following the approach of (Rathinasamy et al., 2014; Yan and Gao 2007; Hu and Si 2016), we test MSES using a set of case studies including stationary and non-stationary synthetic data. The details of the case studies and the wavelet power spectra

are given in Table 1 and Figure 4, respectively.

Case I: A single synthetic stationary time series (S) is generated and contaminated with two random white noise time series. Two sub-cases with different noise-to-signal ratios are investigated (Table 1). This case allows understanding how the synchronization between two series is affected by the presence of noise or high frequency features. For climate variables such situations can emerge when two signals originate from the same parent source or mechanism (e.g. identical large-scale

climatic mode, identical storm tracks) but get covered by high frequency fluctuations arising from local features.

Case II (a): Here we generate two stationary signals consisting of partly shared long-term oscillations and autoregressive (AR1) noise $S_t$ (see Tab. 1). The long-term oscillations $y1, y2, y3$ and $y4$ have periods of $16, 32, 64$ and $128$ units, respectively (Figure 4, Panel I). The purpose of case II (a) is to test the ability of MSES to identify synchronization in

processes which originate from different parent sources or different mechanisms (e.g. two different climatic process, different storm tracks) but have some common features (y1 and y4) at coarser scales.

Case II (b) presents two signals having no common features across all scales. Feature $y2$ in the signal $S1$ and feature y4 in the signal S2 represent a long-term oscillation of period 32 and 128 units, respectively. The idea is to investigate the possibility of overprediction of synchronization if we analyse at one scale only.

Case III: Here, MSES is tested using non-stationary signals generated as proposed by (Yan and Gao 2007; Hu and Si 2016), The signal encompasses five cosine waves (z1 to z5), whereas the square root of the location term results in a gradual change in frequency. Two combinations are generated of which case III(a) investigates the ability of MSES to deal with non-stationarity signals. Case III(b) examines the capability of MSES to capture processes emerging at lower scales (in this case at scale 5 and 6) in the presence of short-lived transient features. For both combinations, the signal is contaminated with white noise.

The time series of case III have features that are often found in climatic and geophysical data, where high-frequency, small-scale processes are superimposed on low-frequency, coarse-scale processes (Hu and Si 2016). Such structures are widespread in time series of seismic signals, turbulence, air temperature, precipitation, hydrologic fluxes or the El Niño Southern Oscillation. They can also be found in spatial data, e.g. in ocean waves, seafloor bathymetry or land surface topography (Hu and Si 2016).

## 3.2 Testing MSES with real-world data

For testing MSES with real-world data, we use precipitation data from stations in Germany (Figure 5). 110 years of daily data, from 1 January 1901 to 31 December 2010, are available from various stations operated by the German Weather Service. Data processing and quality control were performed according to (Österle et al., 2006).

Case IV: We use daily rainfall data from the three stations Kahl/Main, Freigericht-Somborn and Hechingen (Station ID: 20009, 20208, and 25005). Considering Kahl/Main (station 1) as reference station, the distance to the two other stations Freigericht-Somborn (station 2) and Hechingen (station 3) are 14.88km and 185.62km, respectively (Figure 5). Rainfall is a point process with large spatial and temporal discontinuities ranging from very weak to strong events within small temporal and spatial scales (Malik et al., 2012). This case explores the ability of MSES, in comparison to ES, to improve the understanding of synchronization given such time series features.

## 4 Results

To evaluate the synchronization between two signals, which can be expressed in terms of events, at multiple scales, we decompose the given time series up to a maximum scales beyond which there is no significant number events. The number of events at a scale is a function of nature of the time series and also the length of the time series under consideration. In most case it was found that the number of events significantly reduced after seven or eight levels of decomposition. We use

the Haar wavelet, as this is one of the simplest but basic mother wavelets. There are several other mother wavelets which could be used for wavelet decomposition, however, it has been demonstrated that the choice of the mother wavelets does not affect the results to a great extent for rainfall (Rathinasamy et al., 2014).

In case I (a) the $noise/signal$ ratio is quite high in the range of 2.7-3 (Table 1), such that the effect of the noise is felt up to scale 7 (Figure 6). Although both signals stem from the same parent source and hence ideally they should possess perfect synchronization (ES~1) at all scales, the ES value at the observational scale ($\lambda = 0$) is moderate (~ 0.7), leading to the interpretation that both signals are only weakly synchronized. In contrast, the proposed MSES approach is able to capture the underlying features (which were hidden in the original signal) at higher scales ($\lambda \geq 1$) by approaching ES values of 1, indicating the actual synchronization between these signals. At the scale $\lambda = 0$ the ES measure is lower because of the heavy noise covering the underlying information. Considering higher scales, the effect of noise is removed through wavelet decomposition, allowing for a more reliable identification of the actual underlying synchronization between the signals. Interestingly, the slight decrease in the ES values at high scale ($\lambda \geq 7$) (Figure 6) might indicate that the essential feature that is responsible for the synchronization at that scale gets removed in form of a detail component (Figure 2). If features are present at a particular scale $\lambda$ and when we go up to the next scale ($\lambda + 1$), those features get removed in the form of the details and essentially the synchronization is lost at the scale $\lambda + 1$.

While repeating the same analysis but with a lower $noise/signal$ ratio (i.e. case I(b)), we find that the effect of noise is almost completely removed after ($\lambda > 3$) and the MSES values remain unaltered because of the same signal structure (Figure 6). These findings confirm that the MSES approach is able to capture the synchronization in the presence of noise. The significance test (Section 2.4) underlines the high level of synchronization as indicated by the quite high ES values (Figure 6). Based on this example we find that the MSES analysis captures the synchronization at multiple scales.

Case II (a) presents a system where synchronization between two signals exists at a common long term frequency (y1 and y4). This is particularly relevant in studying the rainfall processes of two different regions, which are governed by different local climatic processes but similar long-term oscillations such as ENSO cycles. The MSES values ($\lambda = 0\ to\ 7$) are smaller than the confidence level except for scales 4 and 7 (Figure 7a). The synchronization emerging at scale 4 ($\lambda = 4$) and scale 7 ($\lambda = 7$) correspond to features present at those scales shown in the wavelet power spectrum (Figure 5, Panel I). The thick contour in the WPS indicates the presence of significant features (at 5% significance level) corresponding to $y1, y2, y3$ and $y4$ (Table 1). In the same figure, the dashed curve represents the cone of influence (COI) of the wavelet analysis. Outside of this region edge effects become more influential. Any peak falling outside the COI has presumably been reduced in magnitude due to zero padding necessary to deal with the finite length of the time series. To test the statistical significance of WPS, a background Fourier spectrum is chosen (Addison 2005; Agarwal et al., 2016).

For case II(b), we would expect that the ES value should be zero or nonsignificant at scale $\lambda = 0$. However, we find that the synchronization between S1 and S2 at scale $\lambda = 0$ is significant (Figure 7b), although there is no common feature by construction (Figure 3, Panel II).

Interestingly, the MSES does not find significant synchronization at any scale ($\lambda > 0$). Moreover, the MSES values become zero after scale 4 because the signals S1 and S2 have no common feature beyond these scales.

As seen clearly, the ES at only one scale over predicts the actual synchronicity between the two series. This behaviour may be due to the integrated effect of all scales and hence some spurious synchronization (although rather small but still significant) is indicated.

Case III(a) is used as an analogue of dynamics and features of natural processes (Table 1). Its WPS (Figure 4 Panel III) shows non-stationary, time-dependent features at higher scales $2 \leq \lambda \leq 6$. ES values at lower scales $\lambda \leq 1$ are below the significance level, revealing that the two signals are not synchronized (Figure 8a). The ES for the signal components of the larger time scales reveals significant synchronization up to scale 6, which is expected because of the common features (scale 2 to scale 6) in S1 and S2. After scale $\lambda = 6$, the MSES value drops below the significance level as the features responsible for synchronization are removed in form of the details component during decomposition. Results from this case show the wavelet's ability in capturing the underlying multiple non-stationarities that are common in both the time series which otherwise go unnoticed using ES at the observation scale.

The similar case III(b) is used to investigate the behaviour of MSES in a scale-emerging process in a non-stationary regime (Table 1). As the wavelet spectrum of the signal reveals, only features at scales 5 and 6 are present (Figure 4 Panel IV). The corresponding MSES values are significant only at those scales (Figure 8b), revealing the synchronization at scales 5 and 6. This case illustrates that MSES reveals only the relevant timescales and does not mix them with the observation scale. In reality, there may be situations where the causative events act only at certain time scales and remain unconnected at other time scales. Under such situations MSES is useful to unravel the relevant scale emerging relationships.

After testing the efficacy of the proposed MSES approach by using some prototypical situations, we apply the approach to real observed rainfall data (Case IV). We find significant ES values between station 1 and station 2 at the scales $\lambda = 1, 5 \,\&\, 7$ (Figure 9a) tracking the features present in the WPS (Figure 9 c, d & e). The significant ES value at observational scale ($\lambda = 0$) might be due to the integrated effect of features present at coarser scales ($\lambda = 1, 5 \,\&\, 7$). In order to emphasize the features present in the data, we use the global wavelet spectrum (Figure 9 f,g&h) which is defined as the time average of the WPS (Agarwal et al., 2016; Mallat 1989).

Applying ES in the traditional way, i.e. analysing only at scale 0, we find synchronization. However, only when we consider multiple scales, we are able to find that the synchronization is the result of high and low frequency components present at scales 1, 5, and 7.

For station 1 and station 3 synchronization is significant at scale 7 ($\lambda = 7$) (Figure 9b). However, evaluating the ES in the traditional way (i.e. $\lambda = 0$) leads to the conclusion that both stations are not significantly synchronized. Here, MSES play a critical role in identifying synchronization at specific temporal scales. Hence, MSES provides further insights into the process, such as low-frequency features that are present and the dominating scales causing the significant synchronization at scale 0.

The results for the real-world case study suggest that proximity of stations (station 1 and station 2) does not necessarily indicate synchronization at all scales. For the stations 1 and 3, which are comparatively far from each other, we find insignificant synchronization at the observational scale. However, considering the scales separately MSES detects significant synchronization at scale 7 as both stations might be sharing some common climatic cycle at this scale.

## 5 Discussion

We have compared our novel MSES method with the traditional ES approach by systematically applying both methods to a range of prototypical situations. For the test cases I and II we find that the ES value at the observation scale is influenced by noise, thereby reducing the ES values of two actually synchronized time series. When using MSES, the synchronization between the two time series can be much better detected even in the presence of strong noise. Another important aspect related to the analysis of these cases is that MSES has the ability to unravel synchronization between two stationary systems at time scales which are not obvious at the observation scale (scale-emerging processes). From these observations, it becomes clear that (i) event synchronization only at a single scale of reference is less robust, and (ii) the dependency measure of two given processes based on ES changes with the time scale depending on the features present in these processes.

Case study III illustrates that for a non-stationary system with synchronization changing over temporal scales, the single-scale ES is not robust. In contrast, MSES uncovers the underlying synchronization clearly. MSES is able to track the scale-emerging processes, scale of dominance in the process, features present.

The real-world case study IV shows that the synchronization between climate time series can differ with temporal scales. The strength of synchronization as a function of temporal scale might result from different dynamics of the underlying processes. MSES has the ability to uncover the scale of dominance in the natural process.

Our series of test cases confirms the importance to apply a multi-scale view in order to investigate the relationship between processes that exist at different time scales. We suggest that investigating synchronization just at a single, i.e. observational, scale, could give limited insight. The proposed extension offers the possibility to decipher synchronization at different time scales, which is important in the case of climate systems where feedbacks and synchronization occur only at certain time scales and are absent at other scales.

## 6 Conclusion

We have proposed a novel method which combines wavelet transforms with event synchronization, thereby allowing to investigate the synchronization between event time series at a range of temporal scales. Using a range of prototypical situations and a real-world case study, we have shown that the proposed methodology is superior compared to the traditional event synchronization method. MSES is able to provide more insight into the interaction between the analysed time series.

Also, the effect of noise and local disturbance can be reduced to a greater extent and the underlying interrelationship becomes more prominent. This is attributed to the fact that wavelet decomposition provides a multi-resolution representation which helps to improve the estimation of synchronization. Another advantage of the proposed approach is its ability to deal with non-stationarity. Wavelets being made on local bases can pick up the non-stationary, transient features of a system thereby improving the estimation of ES. Finally, it can be concluded that the proposed method is more robust and reliable than the traditional event synchronization in estimating the relationship between two processes.

## Competing interests

The authors declare that they have no conflict of interest.

## Acknowledgement

This research was funded by Deutsche Forschungsgemeinschaft (DFG) (GRK 2043/1) within the graduate research training group Natural risk in a changing world (NatRiskChange) at the University of Potsdam (http://www.uni-potsdam.de/natriskchange). The third author acknowledges the research funding from Inspire Faculty Award, Department of Science and Technology, India for carrying out the research. Also, we gratefully acknowledge the provision of precipitation data by the German Weather Service.

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

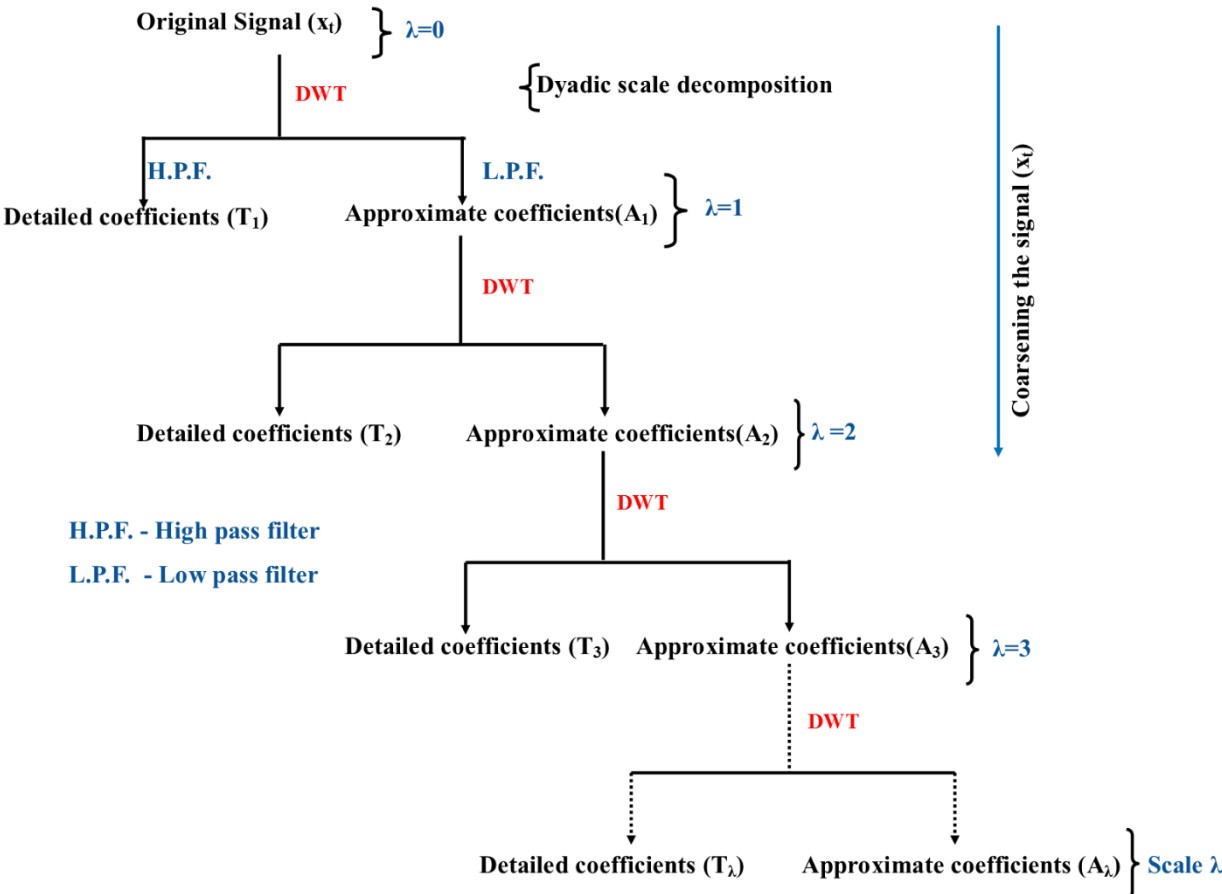

**Figure 1 Schematic showing the decomposition tree for signal $X_t$ using DWT.**

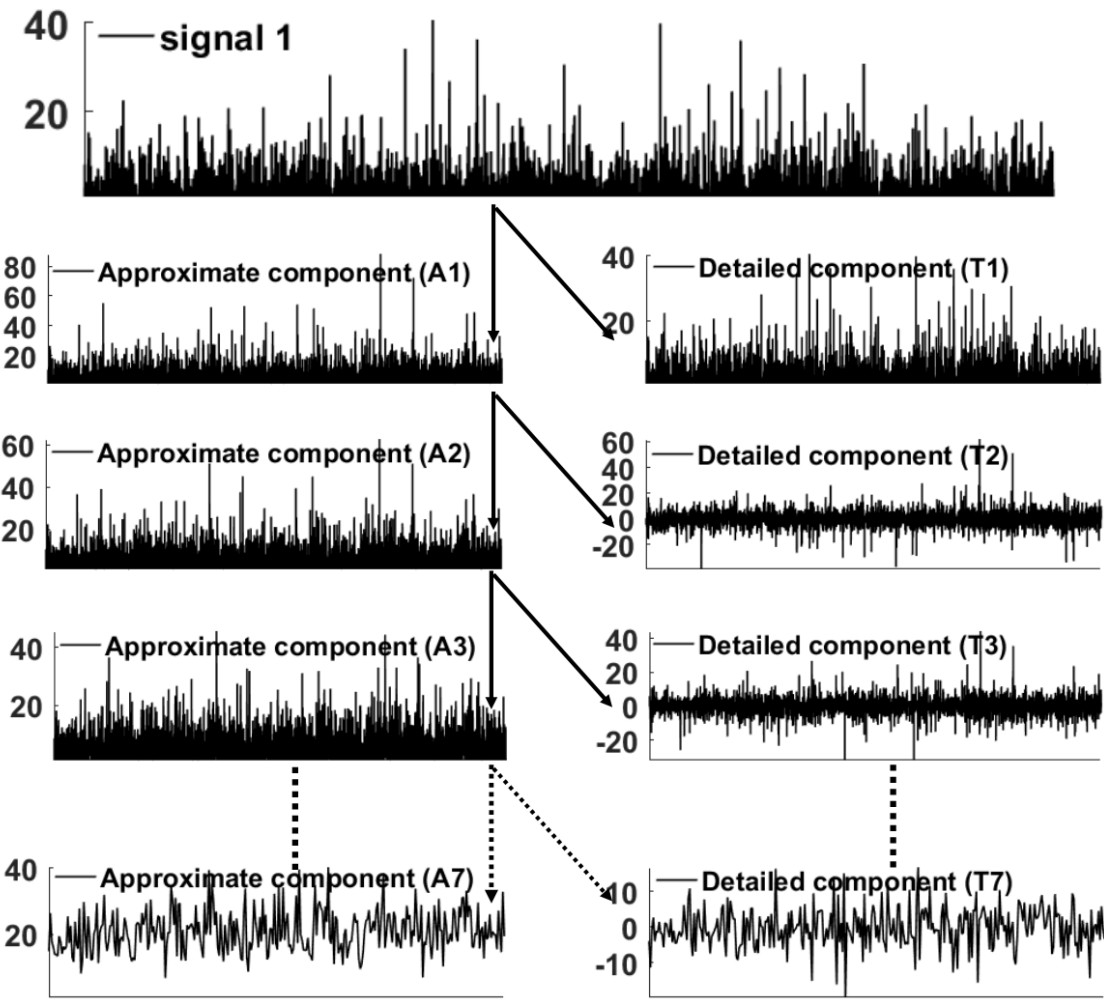

**Figure 2 Scheme of Multi-Scale decomposition of signal using discrete wavelet transformation (DWT). The relationship between signal, approximate component and detailed component is shown.**

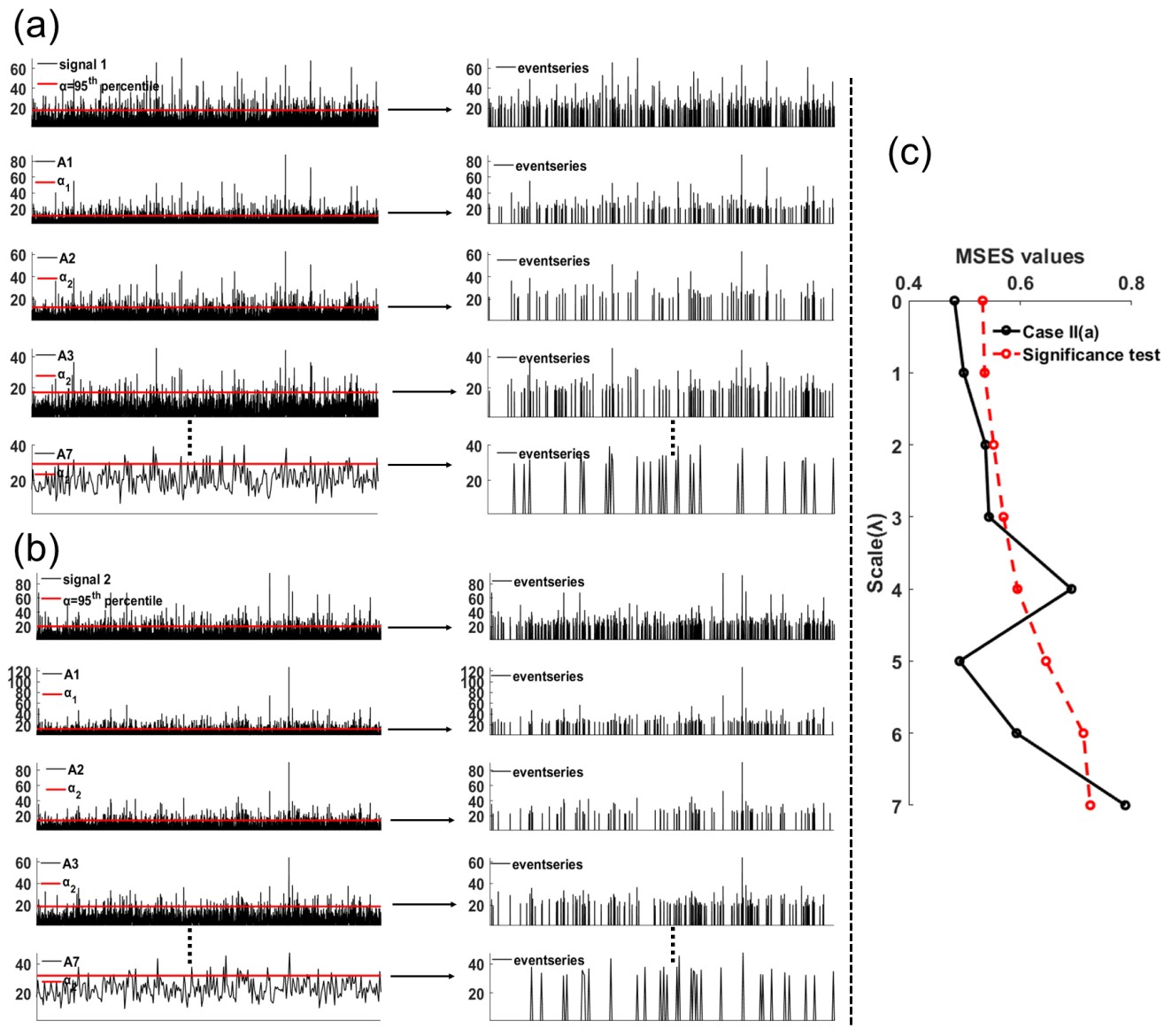

**Figure 3 Multiscale event synchronization (MSES) stepwise methodology. (a) Signal 1 and its decomposed component along with corresponding event series after applying the (95th percentile) threshold. (b) Same for signal 2. (c) Event synchronization values corresponding to each scale.**

**Panel I : for case II(a)**

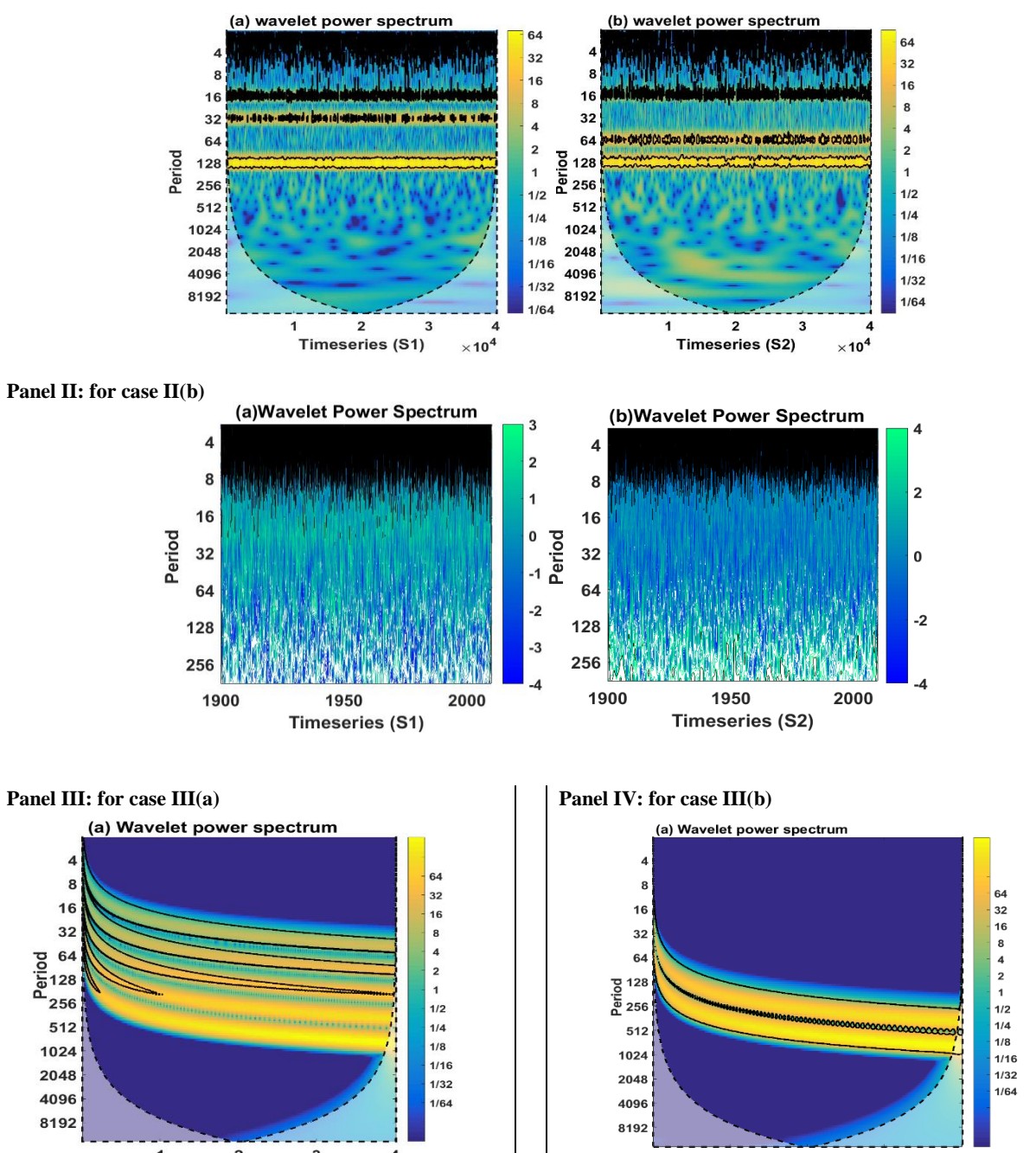

**Figure 4 Wavelet Power Spectra (WPS) of the test signals (Tab. 1). Panel I: original signal S1 (left) and S2 (right) respectively for case II(a); Panel II: original signal S1 (left) and S2 (right) respectively for case II(b); Panel III: original signal S1 for case III(a); Panel IV: original signal S1 for case III(b);**

In all the panels, the y-axis represents the corresponding Fourier period = $2^{\lambda}$ .

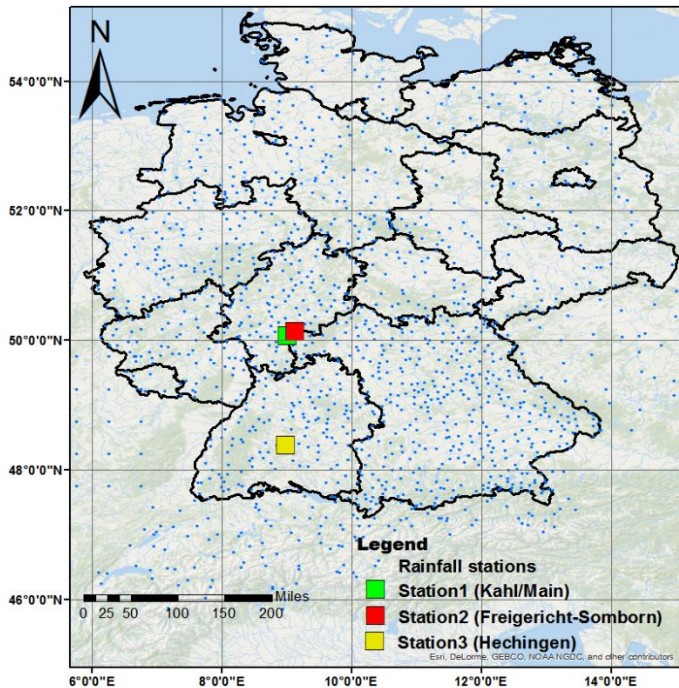

**Figure 5 Geographical location of rainfall stations considered in case study IV**

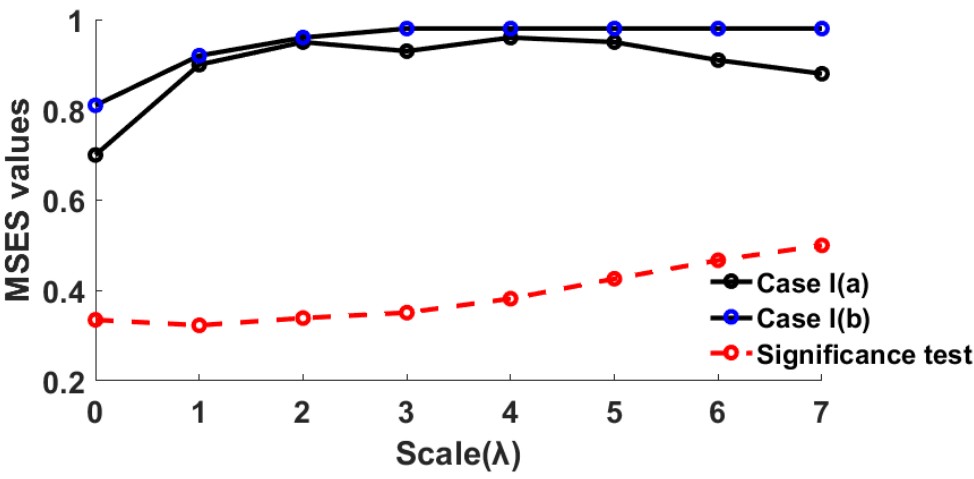

5   **Figure 6 MSES values for case I (a) and (b) including significance test values for significance level of 1%. The value at scale 0 is equal to the single-scale ES analysis.**

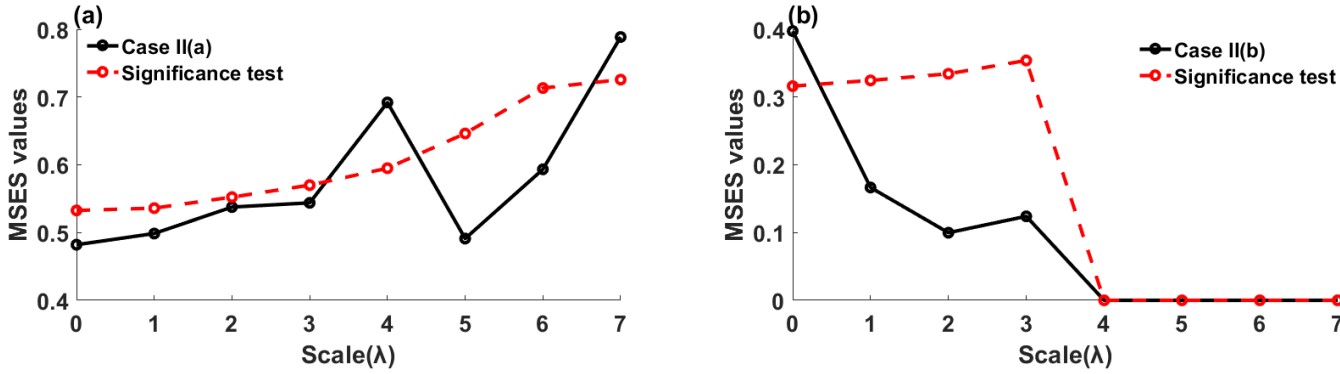

**Figure 7 (a) and (b) MSES and significance level (1%) values at different scales for case II(a) and case II(b). The value at scale 0 is equal to the single-scale ES analysis.**

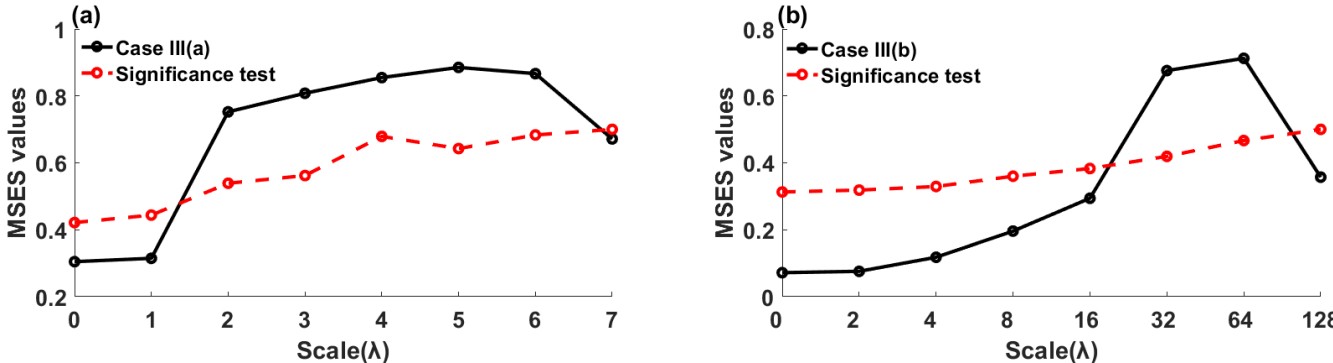

**Figure 8 (a) and (b) MSES values and significance level (1%) at different scales for case III(a) and III(b). The value at scale 0 is equal to the single-scale ES analysis.**

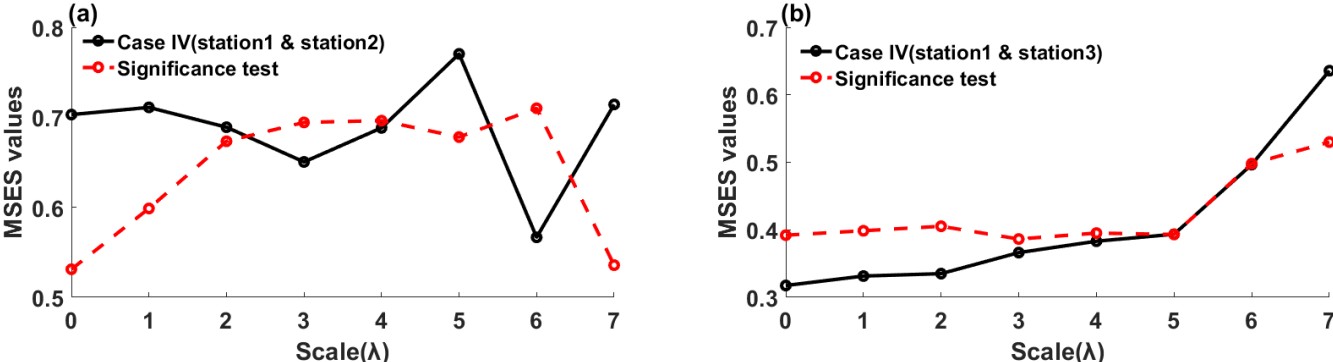

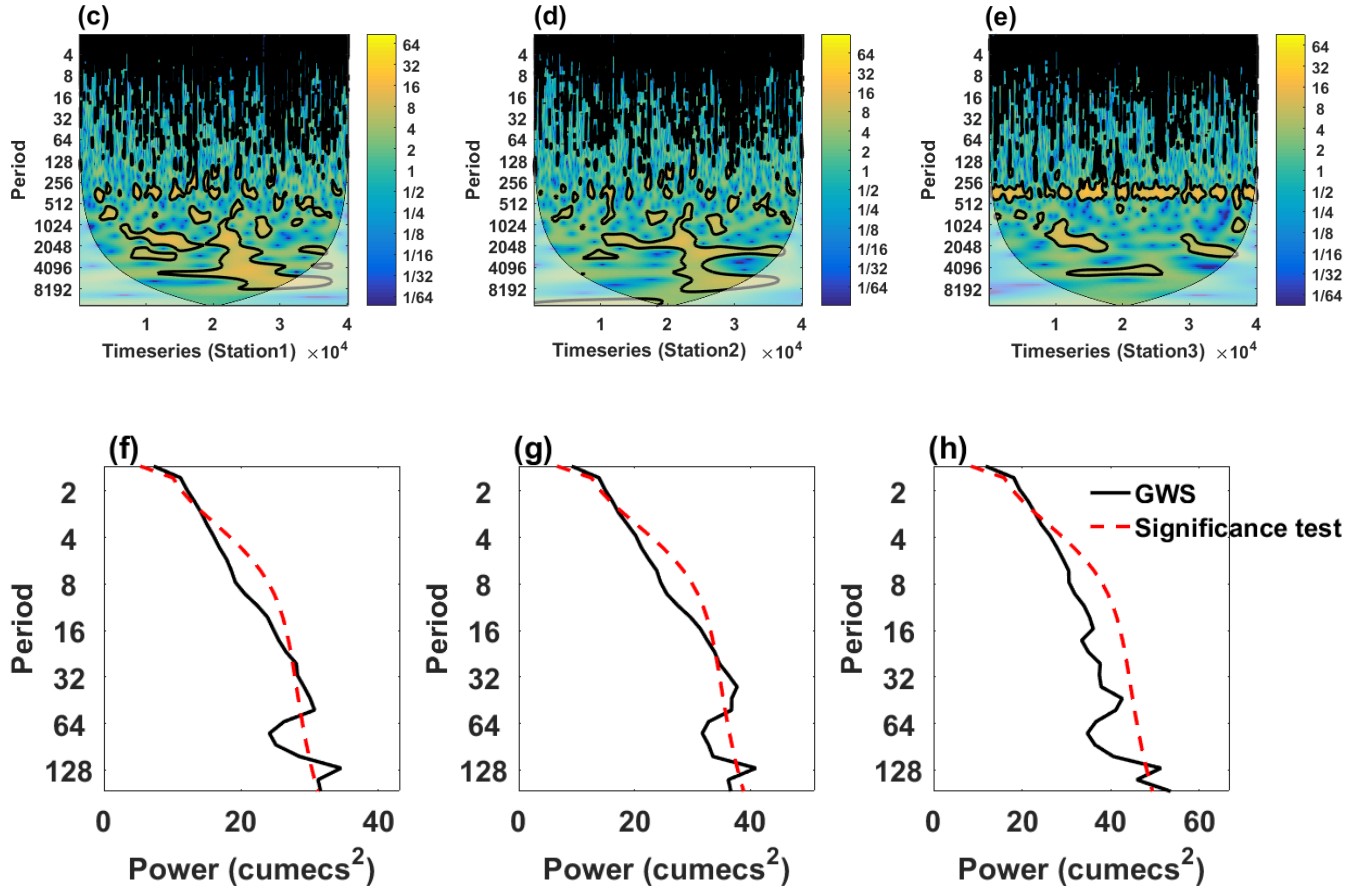

**Figure 9 (a) and (b)** MSES and significance level (1%) values at various scales for station 1 and 2 and station 1 and 3, respectively; (c), (d) and (e) WPS of precipitation of station 1(c), 2(d) and 3(e) (Station ID: 20009, 20208, 25005), respectively; (f), (g) and (h) global wavelet spectrum of the same stations. In the fig. (c)- (h) the y-axis represents the corresponding Fourier period = $2^\lambda$ .

**Table 1: Details of synthetic test cases**

| Case | Mathematical expression | Other details | References and figures |
|------|------------------------|---------------|------------------------|
| I (a) | Sinusoidal stationary signal $S1 = S + \text{Strong noise}_1$ | $S = \sin((2\pi t)/50) + \cos((2\pi t)/60)\}$ <br> $\dfrac{Noise_1}{signal} \sim 2.8$ | Rathinasamy et al., 2014 |
| I(b) | Sinusoidal stationary signal $S2 = S + \text{Weak noise}_2$ | $\dfrac{Noise_2}{signal} \sim .5$ | Rathinasamy et al., 2014 |
| II (a) | Stationary signal (S1&S2) <br> $S1 = S_{t_1} + y1 + y2 + y4$ ; <br> $S2 = S_{t_2} + y1 + y3 + y4$ | Two AR1 process $S_t = \emptyset S_{t-1} + \epsilon_t$ <br> $\varepsilon_t = uncorrelated\ random\ noise$ <br> Parameter $\{\emptyset_1 = .60;\ \emptyset_2 = .70\}$ | Yan and Gao, 2007; Hu and Si, 2016 <br> Figure 3: Panel I |
| II (b) | Stationary dataset (S1&S2) <br> $S1 = y2 + S_{t1}$ <br> $S2 = y4 + S_{t2}$ | $y1 = \sin\left(\dfrac{2\pi t}{16}\right); y2 = \sin\left(\dfrac{2\pi t}{32}\right);$ <br> $y3 = \sin\left(\dfrac{2\pi t}{64}\right); y4 = \sin\left(\dfrac{2\pi t}{128}\right);$ <br> Where t=1,2,3,.....40177 | Yan and Gao, 2007; Hu and Si, 2016 <br> Figure 3: Panel II |
| III (a) | Non-stationary dataset <br> $S1 = z1 + z2 + z3 + z4 + z5$ <br> $S2 = S1 + random\ noise$ $(uncorrelated)$ <br> $\dfrac{noise}{signal} \sim 2.781;$ <br> Where t=1,2,3,.....40177 | $Z1 = \cos\left(500\pi \left(\dfrac{t}{1000}\right)^{.5}\right), Z2 = \cos\left(250\pi \left(\dfrac{t}{1000}\right)^{.5}\right),$ <br> $Z3 = \cos\left(125\pi \left(\dfrac{t}{1000}\right)^{.5}\right), Z4 = \cos\left(62.5\pi \left(\dfrac{t}{1000}\right)^{.5}\right),$ <br> $Z5 = \cos\left(31.25\pi \left(\dfrac{t}{1000}\right)^{.5}\right),$ | Yan and Gao, 2007; Hu and Si, 2016 <br> Figure 3: Panel III |
| III (b) | Non-stationary dataset <br> $S1 = z4 + z5$ <br> $S2 = S1 + random\ noise$ $(uncorrelated)$ <br> $\dfrac{noise}{signal} \sim 21.5664$ ; Where t=1,2,3,.....40177 | $Z4 = \cos\left(62.5\pi \left(\dfrac{t}{1000}\right)^{.5}\right), Z5 = \cos\left(31.25\pi \left(\dfrac{t}{1000}\right)^{.5}\right),$ | Yan and Gao, 2007; Hu and Si, 2016 <br> Figure 3: Panel IV |