# Peer review of "Multi-scale event synchronization analysis for unravelling climate processes: A wavelet-based approach"

_Nonlinear Processes in Geophysics, 2017_

## Referee Comment (RC1) · Anonymous Referee #1 · 11 Jul 2017

General comments

The manuscript proposes a new method for the detection of synchronization between event time series based on the discrete wavelet transform. The approach is straightforward being based on the already established event synchronization method but applied on a scale-by-scale basis to wavelet components instead of the original signal. The idea is sound, however I think that the manuscript needs to be improved, particularly in what concerns the presentation and better explanation of the actual results. In the present form the performance of the method is difficult to assess and thus the claims of the manuscript are not well supported. One of my main concerns is the type of

synthetic test cases performed, since the connection between the smooth and regular signals used for illustrating the method and actual event time series is not clear to me. Is the approach identifying time series correlation or synchronization between events? I think that showing examples of actual event time series (instead of only the wavelet power spectrum) could be helpful. In my opinion the procedure should be shown in more detail at least for one case study: the time series, the corresponding discrete wavelet decomposition, and the results of the quantity Q.

Specific comments

Figure 1: I understand the idea of illustrating the whole proposed procedure using Figure 1, but the figure is only very briefly mentioned in the introduction and not even the caption gives much further detail, for example a short description of a) and b) panels is not given. The quality of the figure itself is also poor.

Eq (4): I think that the exponent of ao in eq (4) should be lambda (and not 2)

Figure 2: Is this figure (from another source, included in the caption) really needed? The quality is very bad (particularly the left side) and doesn't really add much new information. Maybe merge Figures 1 and 2??

Page 5, line 10: Eq(11) is redundant

Figure 3: the wavelet power spectrum is from a continuous wavelet? For consistency shouldn't be shown instead the spectrum based on the discrete wavelet transform? the connection between the periods in Fig 3 and the scale lambda in Figs 5 to 10 should be indicated.

---

## Referee Comment (RC2) · Anonymous Referee #2 · 3 Aug 2017

General comments In my opinion the manuscript is interesting and well written. The proposed approach is novel and original because it provides a new view in the use of ES method, increasing potentiality of this method in the investigation of climate processes. I also think that the subject of the manuscript is interesting for the Jnp's readers. For this reason I recommend the manuscript for the publication on JNP. I just would like suggest to the authors few modifications in order to make more accessible the subject also to readers who are not familiar with Wavelet and Event synchronization approaches. Although these suggestions have been put also into the comments of the pdf file in attachment, I summarize them in the following: a) The description of the methodological link between Multiresolution decomposition of

signals and Event synchronization could be explained in more detail. Section 2.1 Discrete wavelet transform, in my opinion should be rewritten more clearly, probably using a less generic formalism and adding an more explicative figure than figure 1 of the multiresolution decomposition; b) How the continuous signals at the different scales have been converted in binary vectors in order to apply ES is just mentioned, but what thresholds have been used and why is never written in the manuscript; c) There are some parts of 'Section 4 Results' that should be developed in more detail, especially , the results concerning non stationary time series IIIa and IIIb. More discussion about the capability of the approach to treat with non-stationary time series and to capture emerging scales would enhance the paper contents. What are the differences between this approach and the multi-wavelet approach by Hu and Si (2016)? Since the papers use the same synthetic time series should be interesting to compare the two different approaches, their advantages and limitations. d) The quality of the figures should be improve: 1. Figure 1 has distorted axes. Font size should be increased. 2. Figure 2 Font size should be increased 3. Figure 5,6,7,8,9 have distorted axes. Eg. They are squeezed or elongated.

Please also note the supplement to this comment:
https://www.nonlin-processes-geophys-discuss.net/npg-2017-19/npg-2017-19-RC2-supplement.pdf

**Supplement:**

[revised manuscript text omitted]

**2.3 Multi-scale event synchronization (MSES)**

[Figure]

10  Next, we propose to combine both approaches (DWT and ES) to analyze synchronization at multiple temporal scales. Recalling eq. 6, the scale-wise approximation at different scales $0, 1, 2, \dots, \lambda$ for any given time series x(t) is given by:

$$x_\lambda = A_{\lambda,q} = \int_{-\infty}^{\infty} x(t)\phi_{\lambda,q}(t)dt \tag{11}$$

where $x_\lambda$ represents the approximation coefficients of signal x(t) at scale $\lambda$. At that scale we extract an event series out of it which is merely a time series that includes only extreme events. However, ES is not limited to this definition of events. It could also be applied to time series which are pure event time series (e.g. heart beats).

[revised manuscript text omitted]

10  converting a non-stationary time series into stationary components at coarser scales.
[Figure]

The similar case III(b) is used to investigate the behaviour of MSES in a scale-emerging process in a non-stationary regime (Table 1). As the wavelet spectrum of the signal reveals, only features at scales 5 and 6 are present (Figure 3 Panel IV). The corresponding MSES values are significant only at those scales (Figure 9), revealing the synchronization at scales 5 and 6. This example demonstrates the potential of MSES to provide additional information for time series with scale-emerging

15  processes.
[Figure]

[revised manuscript text omitted]

**Figures**

[Figure]

**Figure 1: Scheme of Multi-Scale Event Synchronization (MSES) analysis**

[Figure]

**Figure 2: Discrete wavelet transformation methodology. Left: detailed and scaling coefficient mechanism; right: dyadic decomposition mechanism (Source: Addison, 2002).**

[Figure]

[Figure]

**Panel I : for case II(a)**

**Panel II: for case II(b)**

**Panel III: for case III(a)**    **Panel IV: for case III(b)**

**Figure 3: Wavelet Power Spectra (WPS) of the test signals (Tab. 1). Panel I: original signal S1 (left) and S2 (right) respectively for case II(a); Panel II: original signal S1 (left) and S2 (right) respectively for case II(b); Panel III: original signal S1 for case III(a); Panel IV: original signal S1 for case III(b);**

[Figure]

[Figure]

[Figure]

**Figure 4: Geographical location of rainfall stations considered in case study IV**

[Figure]

.

**Figure 5: MSES values for case I (a) and (b) including significance test values for significance level of 1%. The value at scale 0 is equal to the single-scale ES analysis.**

[Figure]

[Figure]

[Figure]

**Figure 6: MSES and significance level (1%) values at different scales for case II(a). The value at scale 0 is equal to the single-scale ES analysis.**

[Figure]

15 **Figure 7: MSES and significance level (1%) at different scales for case II(b). The value at scale 0 is equal to the single-scale ES analysis.**

[Figure]

[Figure]

[Figure]

**Figure 8: MSES values and significance level (1%) at different scales for case III(a). The value at scale 0 is equal to the single-scale ES analysis.**

[Figure]

**Figure 9: MSES values and significance level (1%) at different scales for case III(b). The value at scale 0 is equal to the single-scale ES analysis.**

[Figure]

[Figure]

**Figure 10: (a) and (b) MSES and significance level (1%) values at various scales for station 1 and 2 and station 1 and 3, respectively; (c), (d) and (e) WPS of precipitation of station 1(c), 2(d) and 3(e) (Station ID: 20009, 20208, 25005), respectively; (f), (g) and (h) global wavelet spectrum of the same stations.**

[Figure]

**Table 1: Details of synthetic test cases**

| Case | Mathematical expression | Other details | References and figures |
|---|---|---|---|
| I (a) | Sinusoidal stationary signal $S1 = S + \text{Strong noise}_1$ | $S = \sin((2\pi t)/50) + \cos((2\pi t)/60)\}$ $\dfrac{Noise_1}{signal} \sim 2.8$ | Rathinasamy et al., 2014 |
| I (b) | Sinusoidal stationary signal $S2 = S + \text{Weak noise}_2$ | $\dfrac{Noise_2}{signal} \sim .5$ | Rathinasamy et al., 2014 |
| II (a) | Stationary signal (S1&S2) $S1 = S_{t_1} + y1 + y2 + y4$ ; $S2 = S_{t_2} + y1 + y3 + y4$ | Two AR1 process $S_t = \emptyset S_{t-1} + \epsilon_t$ $\epsilon_t = uncorrelated\ random\ noise$ Parameter $\{\emptyset_1 = .60;\ \emptyset_2 = .70\}$ | Yan and Gao, 2007; Hu and Si, 2016 Figure 3: Panel I |
| II (b) | Stationary dataset (S1&S2) $S1 = y2 + S_{t1}$ $S2 = y4 + S_{t2}$ | $y1 = \sin\left(\dfrac{2\pi t}{16}\right); y2 = \sin\left(\dfrac{2\pi t}{32}\right);$ $y3 = \sin\left(\dfrac{2\pi t}{64}\right); y4 = \sin\left(\dfrac{2\pi t}{128}\right);$ Where t=1,2,3,…..40177 | Yan and Gao, 2007; Hu and Si, 2016 Figure 3: Panel II |
| III (a) | Non-stationary dataset $S1 = z1 + z2 + z3 + z4 + z5$ $S2 = S1 + random\ noise\ (uncorrelated)$ $\dfrac{noise}{signal} \sim 2.781;$ Where t=1,2,3,…..40177 | $Z1 = \cos\left(500\pi\left(\dfrac{t}{1000}\right)^{.5}\right), Z2 = \cos\left(250\pi\left(\dfrac{t}{1000}\right)^{.5}\right),$ $Z3 = \cos\left(125\pi\left(\dfrac{t}{1000}\right)^{.5}\right), Z4 = \cos\left(62.5\pi\left(\dfrac{t}{1000}\right)^{.5}\right),$ $Z5 = \cos\left(31.25\pi\left(\dfrac{t}{1000}\right)^{.5}\right),$ | Yan and Gao, 2007; Hu and Si, 2016 Figure 3: Panel III |
| III (b) | Non-stationary dataset $S1 = z4 + z5$ $S2 = S1 + random\ noise\ (uncorrelated)$ $\dfrac{noise}{signal} \sim 21.5664$ ; Where t=1,2,3,…..40177 | $Z4 = \cos\left(62.5\pi\left(\dfrac{t}{1000}\right)^{.5}\right), Z5 = \cos\left(31.25\pi\left(\dfrac{t}{1000}\right)^{.5}\right),$ | Yan and Gao, 2007; Hu and Si, 2016 Figure 3: Panel IV |

---

## Author Comment (AC1) · 1 Sep 2017

**Reviewer #1**

**Reply to the Review Comments**

We would like to thank the reviewers for reviewing our manuscript, and for their constructive comments and useful suggestions for further improvement. We have revised our manuscript, taking into consideration all the review comments. During the revision, we have also made other changes, facilitated by a fresh reading. Here, we respond to the specific review comments. In what follows, Page numbers and Line Numbers correspond to those in the clean version.

General comments:

**Reviewer 1:**
The manuscript proposes a new method for the detection of synchronization between event time series based on the discrete wavelet transform. The approach is straightforward being based on the already established event synchronization method but applied on a scale-by-scale basis to wavelet components instead of the original signal. The idea is sound, however, I think that the manuscript needs to be improved, particularly in what concerns the presentation and better explanation of the actual results. In the present form the performance of the method is difficult to assess and thus the claims of the manuscript are not well supported.
**Author's Response:**
We thank the reviewer for the constructive summary of our manuscript and also for his/her critical and supportive suggestions.

**Reviewer 1:**
One of my main concerns is the type of synthetic test cases performed since the connection between the smooth and regular signals used for illustrating the method and actual event time series is not clear to me. Is the approach identifying time series correlation or synchronization between events? I think that showing examples of actual event time series (instead of only the wavelet power spectrum) could be helpful. In my opinion, the procedure should be shown in more detail at least for one case study: the time series, the corresponding discrete wavelet decomposition, and the results of the quantity Q.
**Author's Response:**
First of all, we apologize for not explaining clearly enough the procedure of constructing the considered case studies.
For testing the applicability of the proposed MSES method, we intend to use event series with events occurring at different scales. For this purpose, we generated timeseries using sine functions, which are, at a first stage, indeed smooth and regular (All the mathematical details are presented in Table 1). However, from those generated timeseries, we created event series by thresholding, resulting in events that occur at the desired timescales. The proposed MSES approach was thus applied on the obtained event series, not on smooth and regular time series.

In the revised version we shown (Fig. 3) the generated time series (signal 1&2), decomposed time series, corresponding event series (after applying threshold) and event synchronization (Q) value at multiscale.

[Figure]

**Figure 3 Multiscale event synchronization (MSES) stepwise methodology. (a) Signal 1 and its decomposed component along with corresponding event series after applying the (95th percentile) threshold. (b) Same for signal 2. (c) Event synchronization values corresponding to each scale.**

**Specific comments**

Figure 1: I understand the idea of illustrating the whole proposed procedure using Figure 1, but the figure is only very briefly mentioned in the introduction and not even the caption gives much further detail, for example, a short description of a) and b) panels is not given. The quality of the figure itself is also poor.
**Author's Response:** Figure 1 is replaced and the procedure is explained in Figure 3.

Eq. (4): I think that the exponent of $a_o$ in eq. (4) should be lambda (and not 2).
**Author's Response:** Absolutely, we have corrected the same (eq.4) in the revised version.

Figure 2: Is this figure (from another source, included in the caption) really needed? The quality is very bad (particularly the left side) and doesn't really add much new information. Maybe merge Figures 1 and 2??

**Author's Response:** We thank the reviewer for his/her comments. The purpose of this figure was to help the readers to understand the wavelet decomposition and the relationship between approximate and detailed component but in the revised version this has been removed.

However, another figure (Fig. 2) showing clearly all the details is introduced (see below).

[Figure]

**Figure 1 Scheme of Multi-Scale decomposition of signal using discrete wavelet transformation (DWT). The relationship between signal, approximate component and detailed component is shown.**

Page 5, line 10: Eq.(11) is redundant.

**Author's Response:** We thank the reviewer for his comments. In the revised version we have removed the redundant eq. 11.

Figure 3: the wavelet power spectrum is from a continuous wavelet? For consistency shouldn't be shown instead of the spectrum based on the discrete wavelet transform? The connection between the periods in Fig 3 and the scale lambda in Figs 5 to 10 should be indicated.

**Author's Response:** We again apologize to the reviewer for the confusion created. The wavelet power spectrum for each signal was plotted with the idea of giving a clear understanding of the features present in the synthetic dataset. CWT is not used in the methodology but only for visualization purpose.

Since the DWT based spectrum is plotted at dyadic scales it may not be as clear as the CWT based plot. Therefore we chose to show the CWT to highlight the feature present in the synthetic cases used. The connection between the scale and lambda is included in the new version of the manuscript which is $period = 2^{scale}$.

---

## Author Comment (AC2) · 1 Sep 2017

**Reviewer #2**

**Reply to the Review Comments**

We would like to thank the reviewers for reviewing our manuscript, and for their constructive comments and useful suggestions for further improvement. We have revised our manuscript, taking into consideration all the review comments. During the revision, we have also made further changes, facilitated by a fresh reading. Here, we respond to the specific review comments. In what follows, Page numbers and Line Numbers correspond to those in the clean version.

General comments:

**Reviewer 2:**

In my opinion, the manuscript is interesting and well written. The proposed approach is novel and original because it provides a new view in the use of ES method, increasing potentiality of this method in the investigation of climate processes. I also think that the subject of the manuscript is interesting for the Jnp's readers. For this reason, I recommend the manuscript for the publication on JNP. I just would like to suggest to the authors few modifications in order to make more accessible the subject also to readers who are not familiar with Wavelet and Event synchronization approaches. Although these suggestions have been put also into the comments of the pdf file in attachment, I summarize them in the following:

Author's Response: We thank the reviewer for his/her overall positive evaluation of the manuscript. All the comments of the reviewer have been responded to and the required changes have been made in the manuscript.

a) The description of the methodological link between Multiresolution decomposition of signals and Event synchronization could be explained in more detail. Section 2.1 Discrete wavelet transform, in my opinion, should be rewritten more clearly, probably using a less generic formalism and adding a more explicative figure than figure 1 of the multiresolution decomposition.

Author's Response: We thank the reviewer for pointing out this critical issue. In the revised version we have given a detailed figure (Fig. 3) explaining the methodological link between multiresolution decomposition of signal and synchronization.